# External factors influence intrinsic differences in Stx2e production by Porcine Shiga Toxin-producing *Escherichia coli* strains

Sander Van hoorde[1]☯, Nick Vereecke[2,3]☯, Daniel Sperling[4], Xiaohua He[5], Emma Vanbeylen[1], Emma Van Denberghe[1], Eric Cox[1]☯, Bert Devriendt[1]☯*

1 Laboratory of Immunology, Department of Translational Physiology, Infectiology and Public Health, Faculty of Veterinary Medicine, Ghent University, Merelbeke, Belgium, 2 Laboratory of Virology, Department of Translational Physiology, Infectiology and Public Health, Faculty of Veterinary Medicine, Ghent University, Merelbeke, Belgium, 3 Pathosense BV, Lier, Belgium, 4 CEVA Santé Animale, Libourne, France, 5 Western Regional Research Centre, United States of America Department of Agriculture, Agricultural Research Service, Albany, California, United States of America

☯ These authors contributed equally to this work.
* b.devriendt@ugent.be

## Abstract

Porcine Shiga toxin-producing *Escherichia coli* (STEC) strains pose significant challenges to the pig industry. The toxins produced by these strains, particularly Shiga toxin subtype 2e (Stx2e), are associated with a range of clinical symptoms such as diarrhoea and oedema disease, which in severe cases result in death. Understanding the factors that influence the production and secretion of Stx2e is crucial to elucidate porcine STEC pathogenesis and to develop effective therapeutic strategies. Therefore, this study aimed to characterize the variability in Stx2e production among different porcine STEC strains and assess the effect of several external factors, including bile acids and antibiotics. Our results highlighted a substantial variation in extracellular Stx2e levels by porcine STEC strains. In addition, bile acids, especially the bile acid deoxycholate, exerted strain-specific effects on these extracellular Stx2e levels. Antibiotics also affected extracellular Stx2e levels with ciprofloxacin and enrofloxacin inducing a substantial increase in toxin production in certain strains. Genome analysis revealed that these strains encode a holin gene downstream of the Stx2e operon. Deleting this holin gene abolished the antibiotic-induced increase in extracellular Stx2e levels, while introducing holin expression in unresponsive strains increased the presence of Stx2e in the extracellular environment. These findings unravel a role for phage holins in Stx2e secretion and highlight the intricate interplay between genetic and environmental factors in regulating Stx2e production in porcine STEC strains. Together, our results offer insights into STEC pathogenesis.

purpose. The work is made available under the Creative Commons CC0 public domain dedication.

**Data availability statement:** The manuscript and supporting information contain all data that were used to build the different figures.

**Funding:** This study was supported by Ceva Santé Animale. The Tecan Spark was acquired via funding from the UGent industrial research fund (F2023/IOF-Equip/099 to BD). NV was funded by a grant from the Flemish Agency for Innovation and Entrepreneurship (Baekeland Mandate HBC.2020.2889). The funders had no role in study design, data collection and analysis, decision to publish, or preparation of the manuscript.

**Competing interests:** We have read the journal's policy and the authors of this manuscript have the following competing interests: NV is a former employee at PathoSense BV. DS is employed at CEVA Animal Health.

## Author summary

Shiga toxin-producing *Escherichia coli* (STEC) can cause serious disease in pigs, including diarrhea and oedema disease, sometimes leading to death. These symptoms are mainly caused by the Shiga toxin Stx2e. However, the factors influencing the production and secretion of this toxin by STEC strains are not fully understood. In this study, we examined various porcine STEC strains and tested how external factors like bile acids and antibiotics affect extracellular toxin levels. We found that both strain differences and environmental conditions strongly influence these extracellular Stx2e levels. The bile acid deoxycholate and the antibiotics ciprofloxacin and enrofloxacin, increased toxin production and release in specific STEC strains. This increased toxin release in response to antibiotics was driven by the presence of a gene encoding a holin, a protein produced by viruses that infect bacteria. Removing this *holin* gene stopped the antibiotic-induced increase in extracellular toxin, while adding it to unresponsive strains increased toxin release. These findings reveal how bacterial genetics and external factors interact to control toxin production, improving our understanding of STEC infections in pigs.

## Introduction

Shiga toxin-producing *Escherichia coli* (STEC), also known as verotoxigenic *E. coli* (VTEC), are Gram-negative, facultative anaerobic bacteria that produce Shiga or Shiga-like toxins. These toxins are encoded by the late genes of a toxin-converting lambdoid bacteriophage integrated into the bacterial chromosome [1,2]. After its discovery in *Shigella dysenteriae*, an almost identical toxin (Shiga toxin 1, Stx1) was identified in several *E. coli* serogroups. Later, a second type of Shiga toxin (Stx2) was discovered, which only shares about 56% homology with Stx1, despite its similar mode of action [3]. Since then, additional subtypes have been described, with the prototypic subtypes being Stx1a and Stx2a and newly discovered variants named in alphabetical order. Recently, the number of Stx variants has further expanded to four subtypes of Stx1 (Stx1a, -c, -d, and -e) and fifteen subtypes of Stx2 (Stx2a to Stx2o) [4].

Shiga toxins are ribosome-inactivating proteins, consisting of one A and five B subunits ($AB_5$), which are non-covalently linked to each other. The A subunit harbours enzymatic activity and consists of two domains, A1 and A2. The B subunits on the other hand enable the binding of the toxin to cells expressing specific glycolipid receptors. These B subunits are organized in a pentamer with a central pore that harbours the C-terminus of the A2 subunit domain, facilitating the linkage of the catalytic and binding moiety [5]. Upon binding to the cell membrane, the toxin is internalized and its A subunit is internally cleaved, leading to the release of the A1 domain into the cytosol. There, the A1 domain exerts its rRNA *N*-glycosidase activity and inhibits protein synthesis by inactivating the 60S ribosomal subunit within the

28S rRNA. Consequently, a ribotoxic stress response is triggered, which results in the activation of pro-inflammatory and pro-apoptotic pathways [1,6].

A wide variety of STEC strains have been detected in mammals, birds, fish, and several insects [7]. Even though most animals only represent asymptomatic carriers, postweaning pigs are susceptible to Stx2e toxicity. Upon ingestion, the bacteria colonize the gastrointestinal tract and adhere to the small intestinal epithelium through F18 fimbriae [8,9]. This attachment facilitates bacterial growth and toxin production, resulting in epithelial cell death and subsequent focal disruption of the intestinal epithelial barrier, contributing to clinical symptoms such as mild to severe diarrhoea [10]. This subsequently promotes entry of the toxin into the bloodstream, leading to bloody diarrhoea as well as oedema disease (ED). The latter is characterized by swelling of the eyelids and other predilection sites, as well as neurological signs related to brain lesions, eventually resulting in death of the piglets [1]. The disease outcome is however highly variable. Despite the high prevalence of STEC, with up to 68% of all pigs having shed STEC at least once before the age of 24 weeks, only 5–10% of the infected piglets develop clinical symptoms with a mortality rate of around 8% [11,12]. Recent data has shown that Stx2e reduces the viability and function of lymphocytes [13]. This could explain why Stx2e vaccination on ED-negative farms improves production parameters as compared to control animals [12,14]. Alternatively, since disease severity has been correlated to blood Stx2e levels, varying Stx2e production and secretion of different STEC strains may also account for this [15]. Indeed, previous research showed that genetically distant STEC strains can vary in their Stx2e production [16–18]. Genetic factors but also external factors may thus play a role in the disease outcome by altering Stx2e production. Toxin synthesis in STEC is regulated through the activation of an integrated bacteriophage that encodes the toxin genes. Toxin synthesis even occurs when the prophage genome is not intact, as is the case in most STEC, and the viral genes are spread throughout the host genome [19]. In the lysogenic state, transcription of most phage genes is silenced by the phage-encoded CI repressor [20]. Upon activating the bacterial SOS response due to stressors or environmental cues, this blockade is lifted by a RecA-mediated repressor cleavage. As such, under the influence of the late phage promoter pR', Shiga toxins are produced. This is the case for all Stx (sub)types, although sequence variation has been observed both in the promotor and the surrounding genomic region [21].

Several environmental factors, including bile acids, catecholamines, and antibiotics, can trigger the bacterial SOS response. Previous research on human enterotoxigenic *E. coli* (ETEC) strains has shown that bile acids and conjugated bile acids induce the colonization of the epithelium. However, their impact on gene expression, particularly those related to epithelial attachment, can vary. For instance, production of some STa variants in ETEC strains, such as STa5, is influenced by bile acids, while other variants, like STa3/4, are unaffected [22]. In enterohemorrhagic *E. coli* (EHEC), bile acids such as glycocholate and deoxycholate were found to activate several stress promoters that are part of the SOS response, but not RecA, while in other research they were shown to reduce the production of Stx2 [23]. Furthermore, while the individual bile acids found in humans, cattle, and pigs are similar, their bile compositions differ in the relative abundance of cholic and deoxycholic acids, as well as in the ratios of glycine- and taurine-conjugated bile acids. For example, porcine bile contains a higher proportion of glycine-conjugated bile acids compared to bovine bile [24–26]. Additionally, bile composition changes during animal growth. In adult pigs, the proportion of deoxycholic acid increases, while the proportion of cholic acid decreases compared to post-weaning piglets [27]. Despite these findings in human *E. coli* strains, the influence of bile acids on Stx2e production by porcine STEC strains remains unexplored.

Catecholamines, such as dopamine, epinephrine, and norepinephrine, are stress hormones involved in the fight-or-flight response. These stress hormones play a significant role in interkingdom signalling, particularly in the interaction between hosts and pathogenic bacteria. Norepinephrine and epinephrine can be sensed by bacteria, leading to changes in their behaviour and virulence. Previous work has shown that (nor)epinephrine can indirectly induce the bacterial SOS response [28]. Moreover, norepinephrine has been shown to trigger growth and Stx production in a limited number of human O157:H7 strains [29]. Since norepinephrine-containing sympathetic nerve terminals are part of the enteric nervous

system, it is hypothesized that during the postweaning period, which induces high levels of stress, catecholamines are released into the gut lumen [28]. Hitherto, it is unknown whether catecholamines affect porcine STEC strains.

Although bacterial infections are often treated with antibiotics, in the case of STEC infections this is contraindicated as antibiotic treatment is associated with an increased likelihood of developing more severe symptoms [30–32]. While some studies suggest this could be due to antibiotic-related cell death resulting in the release of toxin, the main theory is that the antibiotic elicits a bacterial SOS response, thereby inducing the lytic cycle of bacteriophages that encode Stx2, thus enhancing toxin production and inducing bacteriolysis [31,33,34]. Research on a limited number of human STEC strains, mainly O157:H7, has shown that antibiotics targeting DNA synthesis and replication, such as mitomycin C and enrofloxacin, can trigger the SOS response in STEC, leading to increased Stx production [33,34]. However, in porcine STEC strains, the *stx2e* genes are encoded on the chromosome, and Stx-converting phages have not been isolated from these strains [15,35]. Information is lacking on the impact of antibiotics on Stx2e production by porcine STEC strains. In addition, not all STEC strains alter their toxin production in the presence of antibiotics that induce the SOS response for yet unknown reasons [18,36].

Thus, this study aimed to understand how genetic factors as well as external factors, such as bile acids, catecholamines, and antibiotics, influence Stx2e production by porcine STEC field strains.

## Materials and methods

### Ethics statement

Animal experiments were reviewed and approved by the Ethical Committee of the Faculties of Veterinary Medicine and Bioscience Engineering at Ghent University, following the Belgian law on animal experimentation (EC2024/038).

### Collection and isolation of wild-type STEC strains

Strains (n = 58) were collected on farms with a previous history of oedema disease (ED) from four different geographical locations (Belgium, the Netherlands, the Czech Republic, and Spain) (S1 Table). Faecal samples were collected by veterinarians using a rectal swab, after which the samples were transferred to our facilities (Merelbeke, Belgium). Isolation of *E. coli* colonies was performed using differential agars, including MacConkey agar (Oxoid) for the isolation of overgrown colonies and their differentiation based on lactose fermenting activity, and Colombia agar (BD) with 5% sheep blood (Bio Trading) for morphological differentiation of the single colonies providing information on their haemolytic activity. Selected colonies of interest were purified and showed growth on both agars, were lactose fermenting and had an off-white, round, and smooth appearance.

### Multiplex-PCR

Colonies of interest were further characterized with a multiplex-PCR using six primer pairs directed against the following virulence factor genes: *faeG, fedA, eltB, estII, estIa,* and *stx2eA* (Table 1) [37]. For each reaction, a master mix was prepared containing 2 µL of DEPC-free water (Abion, #AM9937), 2.5 µL of 10x PCR buffer (Sigma-Aldrich, #D9307), 1 µL of Jumpstart *Taq* DNA polymerase (Sigma-Aldrich, #D9307), 0.5 µM of each forward and reverse primer (IDT), and 0.25 mM of a dNTP stock solution (Roche, #1169004001). To ensure consistency across all samples, the master mix was prepared for the total number of reactions, including controls. After preparation, 18 µL of the master mix was dispensed into each PCR tube. Subsequently, 2 µL of DNA extract or control DNA (F107/86: F18+, Stx2e+; GIS26: F4+, LT+, STa+, STb+) was added to the corresponding PCR tubes for a total volume of 20 µL. The PCR protocol consisted of an initial pre-denaturation step at 90°C for 3 min, followed by 30 cycles of amplification. Each cycle included denaturation at 90°C for 1 min, annealing at 55°C for 1 min with an incremental increase of 3 sec per cycle, and primer extension at 70°C for 2 min. The program concluded with a final extension at 70°C for 10 min. A glycerol stock (50% glycerol, 50% LB) of each

**Table 1. Primer pairs used in multiplex-PCR – All six primers are listed according to their target gene, virulence factor, oligonucleotide sequences (5'->3' direction), and amplicon size (bp).**

| Name | Virulence factor | Oligonucleotide sequence of forward and reverse primes (5' ->3') | Amplicon size (bp) |
|------|-----------------|---------------------------------------------------------------|--------------------|
| estII | STb | TGCCTATGCATCTACACAAT | 113 |
| | | CTCGAGCATGATTACATCTTA | |
| estIa | Sta | CAACTGAATCACTACGTGACC | 158 |
| | | TTAATCAGGCAGTACCATCAGG | |
| eltB | LT | GGGTCTCCTACATTCTGGATG | 272 |
| | | TGGCCTCAGTGAACATGACA | |
| fedA | F18 | TGGTAGAGACAGGTGAGACATA | 313 |
| | | GAAATGTCTCTGACTCTGTAAC | |
| faeG | F4 | GATATGGATTTAACTGGAGGA | 499 |
| | | GTTAGGTGACCAGAGACGAC | |
| stx2eA | Stx2 | AATGATGACAGCGACGATGT | 733 |
| | | TCTGACGTCGTGGTAGTGACTC | |

strain carrying the *stx2e* gene was stored at -80°C. All strains were Stx2e positive except for strain 1705, which is included as a negative control throughout the manuscript.

## Antimicrobial resistance profiling

Each strain was submitted to Dierengezondheidszorg Vlaanderen (DGZ) for antimicrobial phenotyping against a panel of 17 different antibiotics, commonly used for gram-negative bacteria [35]. These results were supplemented with a more extensive minimum inhibitory concentration (MIC) assay to assess the susceptibility to tetracycline, erythromycin, amoxicillin, ciprofloxacin, and enrofloxacin, ranging from 0.016 to 256 µg/mL and 0.002 to 32 µg/mL for the first three and latter two, respectively. Starting from a glycerol stock, each strain was cultured on Luria-Bertani (LB) agar overnight at 37°C, and a single colony was transferred to 5 mL of Brain Heart Infusion (BHI) broth. After overnight incubation at 37°C and 180 rpm, the strains were adjusted to $OD_{600} = 0.5$, and a swab was used to create a spread plate on Mueller Hinton Agar (MHA, BD). Subsequently, the E-test strip (bioMérieux) was applied to the plate and incubated overnight at 37°C. The following day, the MIC value was determined as the lowest antibiotic concentration at which no bacterial growth was observed (S2 Table).

## Growth conditions for Stx2e production

Bacteria were grown from a glycerol stock overnight on Colombia agar (BD) with 5% sheep blood (Bio Trading) at 37°C. Starting cultures were prepared by transferring a single colony from the agar plate to 5 mL of LB medium and incubated overnight at 37°C (180 rpm). Cultures were standardized to $OD_{600} = 1$ and diluted 1:49 into a final volume of 5 mL LB or LB supplemented with compounds before their overnight growth at 37°C (180 rpm). Tested compounds included an 0.3% bile acid mixture from bovine (Oxoid) and porcine (Sigma) sources, 0.3% raw bile extracted from the gallbladder of three 6-week-old piglets at slaughter (bile was extracted with a sterile needle and syringe, filter sterilized (0.22 µm) and pooled), 0.15% cholate (Sigma), 0.15% glycocholate (Sigma), 0.15% deoxycholate (Sigma), 50 µM norepinephrine (Sigma) and 50 µM epinephrine (Sigma). Concentrations were determined based on previous research [22] and on preliminary data in which concentrations ranging from 0.15-2% were tested on a selected number of strains. For antibiotic supplementation, amoxicillin, tetracycline, erythromycin, and ciprofloxacin (Sigma) were supplemented at 1/4th MIC. This is the concentration used in previous research for which the most significant effect was observed [33]. All compounds were either dissolved directly in LB medium or PBS, after which they were filter-sterilized (0.22 µm). The $OD_{600}$ was measured again

and standardized to $OD_{600} = 1$ upon overnight culture. To assess extracellular Stx2e levels, standardized bacterial culture (1 mL) was centrifuged at 10,000 x*g* for 5 min, after which supernatant was collected and stored at -20°C. To assess total Stx2e toxin (intra- and extracellular levels), standardised bacterial culture (1 mL) was incubated with 0.1 mM polymyxin B (Sigma, #Y0000355) for 4 h at 37°C to lyse the cells, followed by centrifugation at 3,000 x*g* for 20 min. The supernatant was collected and stored at -20°C.

## Stx2e ELISA

To detect Stx2e production and secretion, an ELISA was performed as previously described with minor modifications [38]. All incubations were performed for 1 h at room temperature unless stated otherwise. In short, a rabbit polyclonal antibody (provided by USDA), directed against the A subunit of all Stx2 subtypes, was diluted to 2 µg/mL in PBS and 100 µL was used to coat a MaxiSorp 96-well plate (ThermoScientific) overnight at 4°C. The plate was washed three times with PBST (PBS + 0.05% Tween20 (Sigma)) and incubated with 200 µL blocking buffer (PBST + 3% BSA (MP Biomedicals)). After washing three times, 100 µL of the bacterial supernatants (see above), a 2-fold serial dilution of purified Stx2e toxin in PBS (positive control and standard, provided by CEVA and described in [39]), and LB medium (negative control) were added in duplicate and incubated. After three washes, 100 µL of a mouse anti-Stx2e monoclonal antibody (provided by USDA) diluted to 1 µg/mL in blocking buffer was added and incubated. After three washes, 100 µL of the secondary poly-clonal rabbit anti-mouse IgG:HRP antibody (Dako, #P0260) diluted 1/1,000 in blocking buffer was added and incubated. Finally, the plate was washed three times, and 100 µL of TMB-solution (Sigma) was added and incubated for 8 min at room temperature. The reaction was stopped by adding 50 µL of 2 M HCl (VWR), and the optical density was measured at 450 nm using a Tecan Spectrafluor/Spark.

## STEC genome analysis

Based on Stx2e positivity using PCR, STEC strains were subjected to whole genome sequencing using long-read nanopore sequencing at PathoSense. Isolation of DNA, library preparation, sequencing, and genome assembly were per-formed as described previously [35]. After genome quality filtering (i.e., genome coverage and completeness [35] and S3 Table), 53 genomes were retained for genetic analyses (strains 1704, 1711, 1712, 1726, and 1727 were excluded). These genome sequences are publicly available through the National Centre for Biotechnology Information (NCBI) under project PRJNA917806 and were analysed to evaluate the genomic relatedness by generating a core genome with Prokka v1.14.5 [40], Roary v.3.13.0 [41] and inferring maximum likelihood (ML) phylogenetic relatedness with IQ-tree2 v1.6.1 with 1,000 ultrafast bootstraps (−bb) [42]. Trees were visualized using iTOL [43]. The Stx2e negative strain 1705 was used as an outgroup for this analysis. Next, the genome context around the Stx2e operon (10,000 bp on each side of the operon) was extracted and analyzed in depth using flanker v0.1.5 [44] and clinker v.0.0.27 [45]. Genomes or genomic segments were annotated using Prokka v1.14.5 [40]. Based on these gene annotations and the roary output, a gene-based genome-wide association study (GWAS) was performed using Scoary v.1.6.16 [46]. For this analysis two different groups were created, based on low (<1 ng/mL; 11 strains), moderate (1–10 ng/mL; 41), or high (>10 ng/mL, 5) Stx2e production. A first analysis, group 1, included both high and moderate Stx2e producers as positive (*i.e.,* 1 in the GWAS metadatafile in S4 Table), whereas the second analysis, group 2, only included the high Stx2e producers as positives (S4 Table). Output files can be found in S5 and S6 Tables. Phaster was used to identify and annotate prophage sequences within bacterial genomes and check for prophage completeness [47,48]. Finally, the presence of genes known to be involved in bacterial responses to bile acids and stress hormones was evaluated using abricate v1.0.1 [49] and run with --mincov and --minid both set to 80% and 60% (S8 and S9 Tables). To study polymorphisms in the Stx2e operon promoter region, upstream regions were extracted and aligned using mafft (v7.525; [50]), followed by promoter prediction using the sigma70pred "scan" function on webs.iiitd.edu.in/raghava/sigma70pred/scan.html [51]. Finally, investigation of different Stx2e and holin configurations was done using the NCBI Prokaryotic Genome Annotation Pipeline (PGAP) annotations as available on NCBI for all of

the submitted genomes. The folding of the putative (phage) holin protein encoded downstream of the *stx2eB* gene was predicted using alphafold2 at default settings [52].

## Small intestine segment perfusion assay

To assess the effects of a bile acid mixture, deoxycholic acid, and enrofloxacin on Stx2e secretion *in situ*, a small intestine segment perfusion (SISP) assay was performed as described previously [53,54] with minor modifications to the anesthesia protocol. Briefly, piglets (n=6) were premedicated by intramuscular injection of 4.4 mg/kg bodyweight tiletamine hydrochloride and zolazepam (Zoletil, Virbac, France) and 4.4 mg/kg bodyweight xylazine (Xylamidor, VetViva Richter, Austria), after which piglets were intubated and maintained under long-term anesthesia as previously described. Piglets (females, 5 weeks old; n=6) were selected to be negative for F18 fimbriae and Stx2e serum antibodies and to be susceptible to F18+ *E. coli* infections using the FUT1 assay [54]. Upon transfer to our animal facilities and an acclimation period of 4 days, piglets were subjected to a SISP assay. Briefly, the abdomen was opened at the linea alba and six small intestinal segments were constructed in the mid-jejunum starting at 200 cm distal to the ligament of Treitz. Segments were 20 cm in length with 5 cm between each segment. These segments retained their vascularization and were cannulated with a rubber tube at the proximal and distal ends to inject and collect fluid, respectively. Intestinal segments were perfused with: 1) 5 mL of perfusion fluid (0.9% NaCl+0.1% glucose) or 2) 5 mL of perfusion fluid containing $5x10^8$ colony forming units (CFU)/mL Stx2e+ bacteria (strain 4080) pre-incubated for 1 h at 37°C with perfusion fluid, 3) 0.3% porcine bile mixture (Sigma), 4) 0.15% deoxycholic acid (Sigma) or 5) enrofloxacin at 1/4th of the MIC. The experiment ran for 6 h and every 15 min, 2 mL perfusion fluid was injected, while the outflow was continuously collected in 50 mL Falcon tubes on ice. Stx2e levels in the outflow samples were evaluated by ELISA as described.

## Construction of holin null mutants

The *holin* gene (putative holin [*Escherichia coli*]; WP_032083190; 130AA) in two STEC strains (4056 and 1717) was removed using the lambda red recombination system, in which an antibiotic resistance gene replaces a target sequence with the use of a red helper plasmid [55]. Therefore, the kanamycin resistance (KanR) cassette of plasmid pET30 was PCR amplified using oligonucleotide primers KanR_FW and KanR_RV (Table 2). These oligonucleotides were 70-mers in which the 50 bases at the 5′ were complementary to regions inside the *holin* gene, followed by 20 bases at the 3′ that flanked the *kanR* cassette open reading frame. Wild-type strains were made electrocompetent by growing them to $OD_{600} \approx 0.6$ in LB, concentrating 100-fold, and washing them three times with ice-cold 10% glycerol [56]. These were then transformed with the Red helper plasmid pKD46 (containing an ampicillin resistance cassette) using electroporation. Here, 20 μL electrocompetent cells (~$10^9$ cells) and 10 ng plasmid were added to a Gene Pulser Cuvette with a 0.2 cm gap (Biorad, #1652086) and placed in a MicroPulser (Biorad, #1652086) set at 2.5 kV. After electroporation, the cells were

**Table 2. Overview of primers used in the deletion and supplementation of holin.**

| Name | Oligonucleotide sequence of forward and reverse primes (5′ -> 3′) |
| --- | --- |
| KanR_FW | TTGTTTTTATGGGCCGCTGGTGGCCCTTTTTTATTTACAGGAGAAAAAGTATGAGCCATATTCAACGGGA |
| KanR_RV | GCGTCAGCACGATAACCGCGCATAACGCCACATTCAGCAATCCGGGAAGGGAAAAACTCATCGAGCATCA |
| pKD_Kan_FW | AAACGTCTTGCTCTAGGCCGCG |
| pKD_Kan_RV | ACGCCACATTCAGCAATCCGGG |
| Holin_FW | TCTAGAATGTCTGAACCCTTGTCCGG |
| Holin_RV | AAGCTTTTACTTAACATTGCCGCCTC |
| pUC57_Holin_FW | GGCGACGGTATTCGGGCTGTTT |
| pUC57_Holin_RV | AGACCGGACACCAGTGATGCGA |

transferred into 1 mL of SOC medium at 30°C, recovered for 1 h, and subsequently plated and incubated overnight on LB agar plates with ampicillin (100 μg/mL) at 30°C. Transformants carrying a Red helper plasmid were selected and grown in 5 mL LB medium with ampicillin (100 μg/mL) and L-arabinose (0.1 M) at 30°C (*i.e.,* pKD46 carries a temperature-sensitive *ori*) to an $OD_{600} \approx 0.6$ and then made electrocompetent again by concentrating the 5 mL 100-fold and washing three times with ice-cold 10% glycerol [56]. Electroporation was done as described before with the exception that 100 ng of PCR product was used instead of plasmid DNA. Cells were then transferred to 1 mL of SOC medium, incubated for 1 h at 37°C, and spread onto agar, containing kanamycin (50 μg/mL) to select for $Kan^R$ transformants. Replacement of the *holin* gene with the $Kan^R$ cassette was verified by PCR using primers pKD-Kan-FW and pKD-Kan-RV (Table 2). One positive clone for each transformation was selected for further study and stored at -80°C in glycerol.

### Holin gene supplementation

The *holin* gene was supplemented in four STEC strains (4055, 4063, 4056 Δholin, and 1717 Δholin) using restriction cloning in an IPTG-inducible plasmid. The *holin* gene of strain 4056 was used as a template and amplified using primers Holin_FW and Holin_RV (Table 2), resulting in the *holin* gene flanked by XbaI and HindIII restriction sites at the 3' and 5', respectively. Subsequently, 1 μg of both the holin amplicon and pUC57 plasmid were separately digested in 5 μL 5x CutSmart buffer (NEB, # B7204), using 1 μL of the XbaI and HindIII restriction enzymes (NEB), diluted with ultrapure water to a total volume of 50 μL, and incubated for 1 h at 37°C. For the ligation reaction, 4 μL 5x rapid ligation buffer (NEB), 1 μL T4 DNA ligase (5 U/μL, NEB), 10–100 ng digested plasmid, amplicon DNA in a 1:3 molar ratio, and ultrapure water to 20 μL was added and incubated for 5 min at 37°C. Both wild-type strains were made electrocompetent by growing them to $OD_{600} \approx 0.6$ in LB, concentrating 100-fold, and washing them three times with ice-cold 10% glycerol [54]. The four strains were then transformed with the pUC57:holin construct using electroporation. Here, electrocompetent cells (20 μL) and 10 ng of plasmid were electroporated as described above. After electroporation, cells were transferred into 1 mL of SOC medium at 37°C, recovered for 1 h, and subsequently plated and incubated overnight on LB agar plates with ampicillin (100 μg/mL) at 37°C. Colonies were selected and screened with PCR using the primer pair pUC57_Holin_FW and pUC57_Holin_RV (Table 2) for the presence of the *holin* gene. One positive clone for each transformation was selected for further study.

### Data analysis

All data were analysed with GraphPad Prism 10. A p-value < 0.05 was considered significant. Homogeneity of variances was tested with a Brown-Forsythe test, while a normal distribution of the data was assessed with Shapiro-Wilk. A paired Student's t-test (piglet bile acids) or one-way ANOVA (individual bile salts, SISP, holin deletion/supplementation) was performed to compare control and test conditions with a normal distribution. A Wilcoxon signed rank test (bile acids) or a Friedman test with Dunn's post hoc analysis was used to analyse data from experiments that did not meet normality assumptions (stress hormones, antibiotics). ELISA data were analysed using DeltaSoft JV software, employing a 4-parameter logistic regression (4PL) model for curve fitting. To determine the Stx2e levels in the perfusion outflows of the SISP assay, values of the control segment were subtracted from the other values.

## Results

### STEC strains differ in their ability to produce Stx2e

Previous research showed that ETEC and STEC strains differ in the levels of secreted toxins. For the STEC strains, data is scarce since only a handful of strains were used in previous research. Here, we evaluated potential differences in Stx2e production and secretion levels by 57 genetically divergent porcine STEC strains (Fig 1). These strains carried the genes encoding the Stx2e toxin, except for strain 1705. Like previous research, the porcine STEC strains

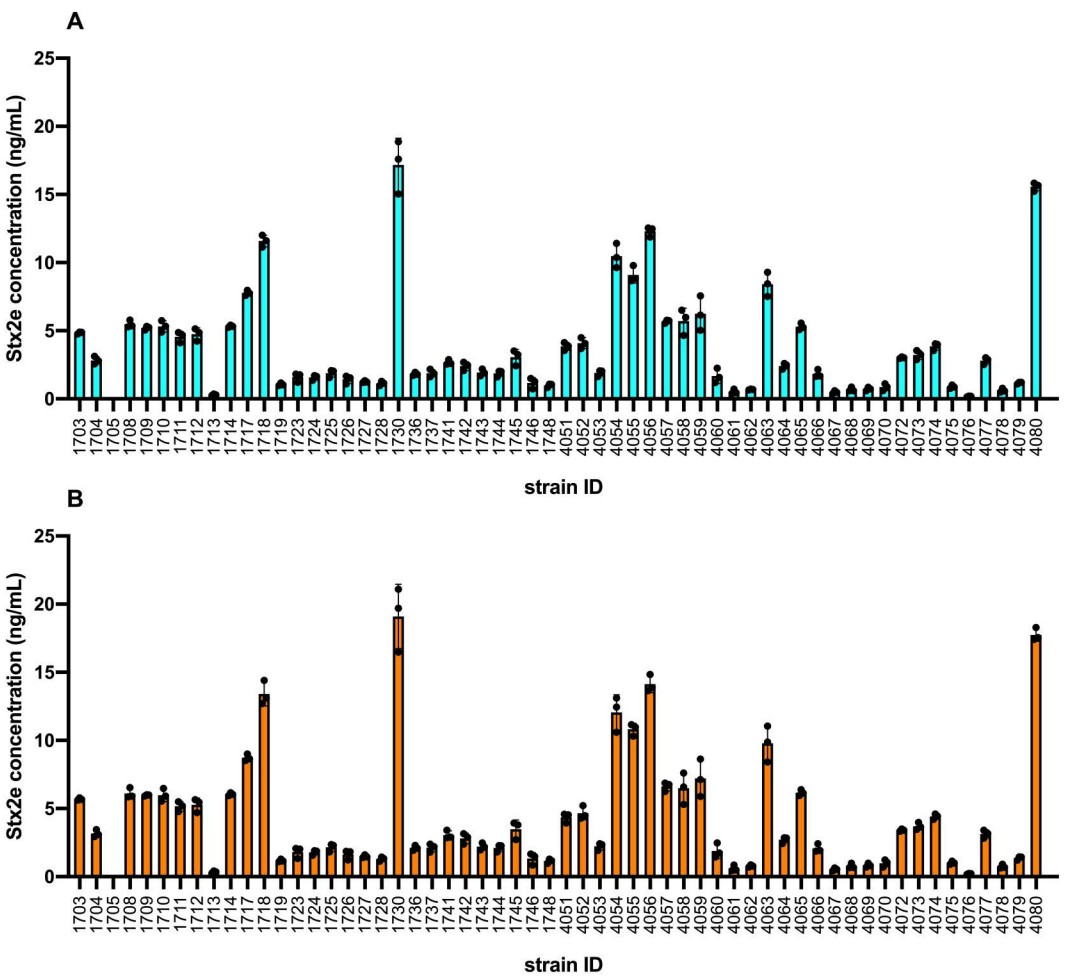

**Fig 1. Inter-strain differences in Stx2e production levels – STEC strains were grown overnight in LB and the secreted and total (secreted and intracellular) Stx2e levels in (A) the bacterial supernatants or (B) after cell lysis (0.1 mM polymyxin B) were determined by ELISA, respectively.** The toxin concentration (ng/mL) is plotted for 58 strains (57 STEC strains and one negative control strain, strain 1705). Each symbol represents an independent experiment (n = 3), and bars denote the geometric mean.

varied considerably in their ability to produce Stx2e (Fig 1B). A 115-fold difference in extracellular Stx2e levels was observed between the highest and lowest secretor. Based on this difference in extracellular Stx2e levels, the strains were divided into three groups: low (<1 ng/mL; 11 strains), moderate (1–10 ng/mL; 41), or high (>10 ng/mL, 5). Our previous research on enterotoxin production by porcine ETEC strains demonstrated a remarkable difference for some strains between produced and secreted levels of the heat-labile enterotoxin (LT), another AB$_5$ toxin [18]. The tested STEC strains however did not show such a difference, as Stx2e production levels matched well with its secretion levels (Fig 1A and S7 Table).

To explain this variation in Stx2e production between strains, the genome of the STEC strains was sequenced [35] and analyzed. Similar to other Shiga toxins, the Stx2e operon was only located on the bacterial chromosome. A detailed prophage characterisation did not identify intact prophages, although prophage genes were found scattered throughout the STEC genomes. Applying a gene-based GWAS to identify genetic mediators associated with low/moderate/high secretors, as defined above, did not result in meaningful results (S3-S6 Tables).

In a further effort to explain the variation in extracellular Stx2e levels between the tested strains, the Stx2e operon and the surrounding genomic region were compared for all tested strains (Fig 2A and 2B). Five non-silent SNPs in the *stx2eA* sequence resulted in several toxin variants, including D175E (T525G; n = 2), T207A (A619G; n = 1), P289L (C866T; n = 1), T296K (C887A; n = 1) S313P (C937T; n = 20) genotypes. The Stx2eA mutations did not correlate with the observed variations in extracellular Stx2e levels. However, for the S313P mutation, a decrease in cytotoxicity was observed. Interestingly, in four strains the *stx2eB* coding sequence was interrupted by a stop codon (E46*; G136T) and resulted in the loss of 42AA at its 3' end (genomic contexts 15 and 16; Fig 2C). All these strains showed a comparable moderate presence of Stx2e in the extracellular environment. When analysing the regions around the Stx2e operon in the 52 STEC strains, a minimum of 16 different genomic contexts could be identified (Fig 2C). In 38 strains (73%) the *stx2e* operon was flanked downstream by two

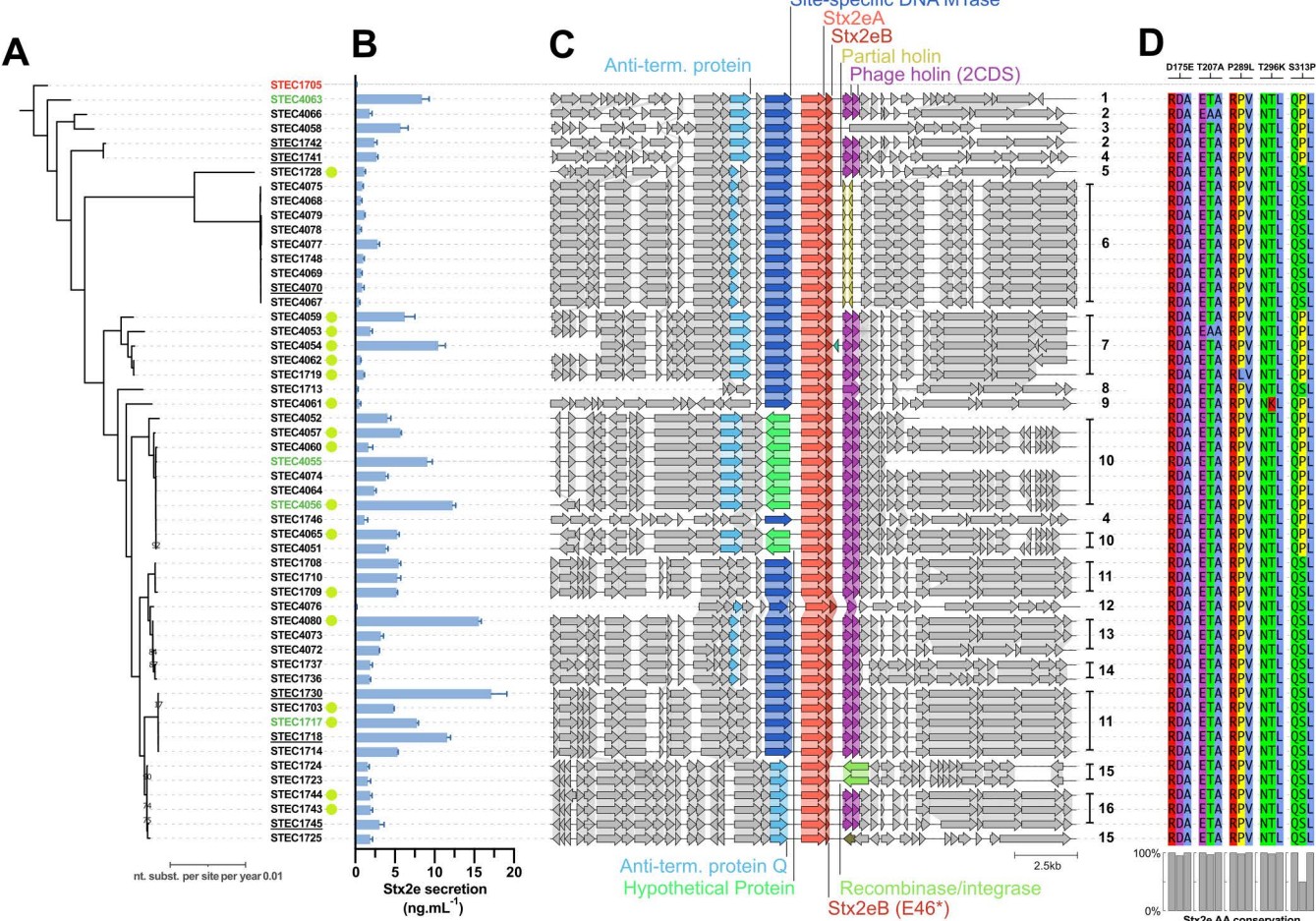

**Fig 2. Inter-strain difference of genetically distinct STEC strains in extracellular Stx2e levels and their Stx2e operon landscape – (A) Maximum-Likelihood (ML) phylogenetic tree of all included STEC strains (n = 52), including strain 1705 as negative control and outgroup (red).** Bootstraps <95 are indicated next to branches. Strains that responded to fluoroquinolones are indicated with a green circle. Underlined strains were used in an in-depth characterization of the impact of external factors (see above) and strains highlighted in green were used in holin deletion experiments. (B) Bar chart of extracellular Stx2e levels (ng/mL) linked to their phylogenetic relatedness. Bars represent the mean, errors bars represent the sd (n = 3). (C) Graphical representation (clinker plot) of the 10,000 bp flanking sequences at both sides of the stx2e operon highlighting immediate up- and downstream annotated coding sequences (CDS). For each differing genomic landscape (*i.e.,* different up- and/or downstream CDS), a clustering/group number was given, ranging from 1-16. MTase = Methyltransferase. Stx2eB (E46*) represent a mutation in the *stx2eB* gene resulting in the introduction of a stop codon. (D) Positional amino acid (AA) conservation of polymorphic sites within the Stx2eA protein. All other positions showed 100% AA conservations.

coding DNA sequences (CDS) encoding a putative holin gene. Genomic context 6 encoded a partial holin gene (n = 9; 17%). In 37 (71%), 9 (17%), and 6 (12%) strains, the *stx2e* operon was flanked upstream by a site-specific DNA-methyltransferase, a hypothetical protein, or a bacteriophage anti-terminator protein Q, respectively. While genomic context 6 comprised most strains with low extracellular Stx2e levels, strains with other genomic contexts showed more variable extracellular Stx2e levels (Fig 2B and 2C). Applying a gene-based GWAS to identify genetic mediators associated with low/medium/high secretors did not result in meaningful results (S3-S6 Tables). As most variation in extracellular Stx2e levels by STEC strains could not be explained by the genomic context around the Stx2e operon, we next assessed whether the promoter region of the Stx2e operon contributed to differential secretion. While some clade-specific mutations were observed, no polymorphisms within the promoter region and beyond could explain the observed differences in Stx2e secretion (S1 Fig).

In a next effort we evaluated whether external factors might influence the intrinsic differences in extracellular Stx2e by porcine STEC strains.

## Bile acids influence extracellular Stx2e levels

Previous research has shown that bile acids can influence toxin production in other bacterial species. In ETEC the production and secretion of LT toxin is downregulated under the influence of bile. To see whether this was also the case for Stx2e in STEC, we first checked whether the genes needed for bile sensing and resistance were present (S9 Table). After confirming the presence of these genes in all strains, except for three strains (4053/4058/4066) that do not encode the *ler* transcriptional regulator, the 57 STEC strains were cultured in the presence of bovine bile acids. Bile acids of bovine origin are most often used to investigate the impact of bile on human bacterial pathogens due to the similarities in the bile composition between both species. While bovine bile acids reduced the presence of Stx2e in the extracellular environment of 22 strains, 23 other strains responded to the presence of bile acids with increased extracellular Stx2e levels (Fig 3A and S1 Table). The other 13 strains did not significantly alter their toxin levels in the presence of bile acids. As mentioned before, the relative content of bile acids changes between species as well as with age. To assess whether bovine and porcine bile acids had a similar impact on the extracellular Stx2e levels, we assessed the impact of porcine bile acids on Stx2e secretion. Porcine bile acids had a similar effect on the extracellular Stx2e levels by the STEC stains as compared to bovine bile acids (Fig 3B). Finally, since age can impact the composition of bile content, we wanted to assess whether bile from younger animals potentially altered Stx2e production. For this, we selected three strains that showed increased or decreased extracellular Stx2e levels in the presence of bovine and adult porcine bile acids (green and red, respectively) and cultured these in the presence of bile collected from recently weaned piglets. As shown in Fig 3C, the results were similar to those obtained with bovine and porcine bile acids.

To determine which components of the bile acid mixtures were responsible for the in- and decrease in extracellular Stx2e levels, the two most abundant bile acids, cholate and deoxycholate, were selected for further testing. Also, glycocholate was used based on findings from previous research on its potency to alter the expression of ETEC virulence genes [22]. Here, the same subset of strains used to evaluate the impact of bile acids from a young piglet on Stx2e production was selected to assess the effect of these individual bile acids on the presence of Stx2e in the extracellular environment. Fig 3D shows that both cholate and glycocholate did not alter extracellular Stx2e levels. Deoxycholate on the other hand decreased extracellular Stx2e levels in strains that reduced toxin levels in response to the bile acid mixture (D1) and increased extracellular Stx2e levels in strains that increased toxin levels in response to the bile acid mixture (D2). Altogether, these data suggest that most porcine STEC strains sense bile acids, especially deoxycholic acid, and in turn alter their Stx2e production.

## Catecholamines do not influence extracellular Stx2e levels

Previous research in EHEC showed that the production and secretion of Stx2 are upregulated under the influence of norepinephrine. To see whether this was also the case for Stx2e in porcine STEC, as well as for epinephrine, we first

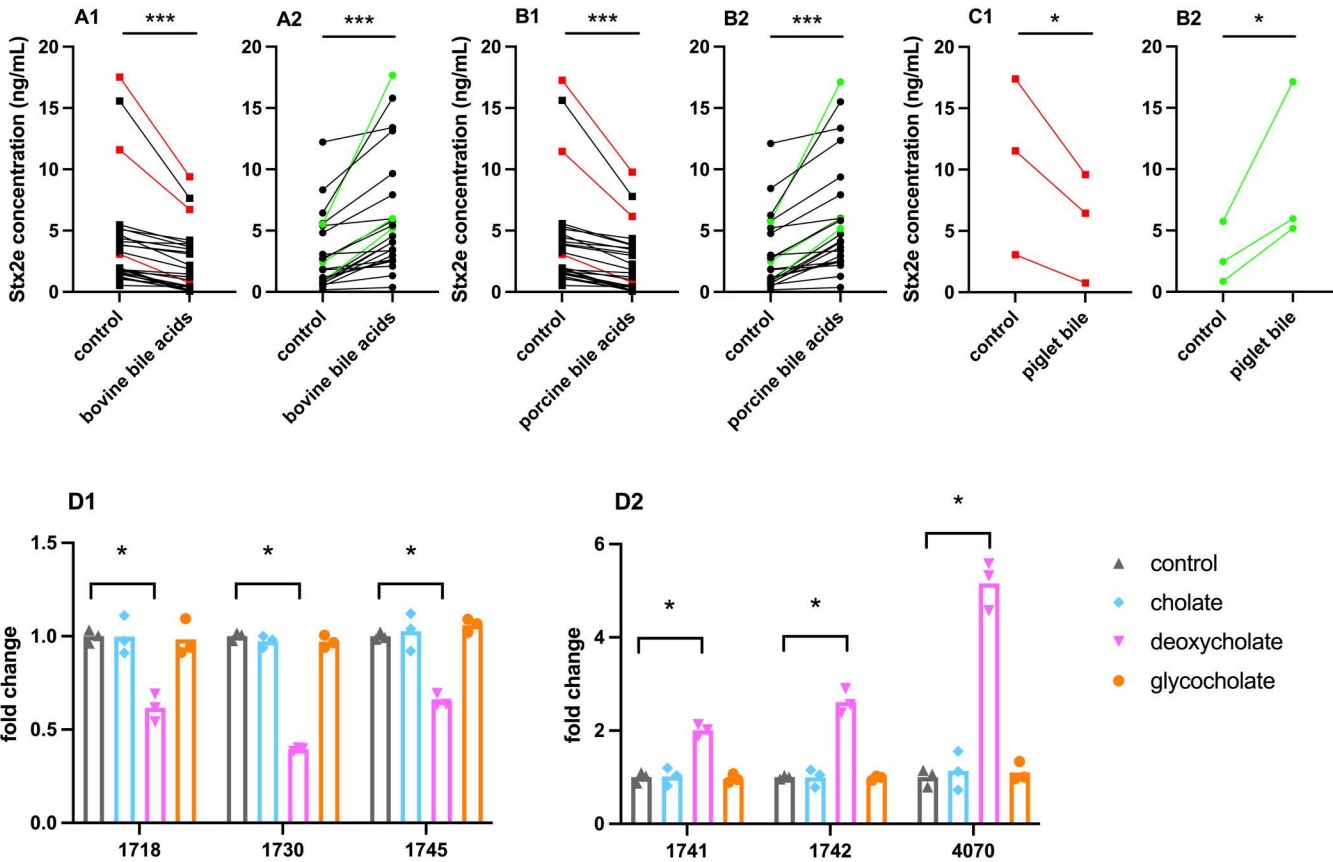

**Fig 3. Influence of bile acids on extracellular Stx2e levels – STEC strains were grown overnight in LB medium with either 0.3% bovine bile (A), 0.3% porcine bile (adult) (B), 0.3% bile obtained from recently weaned piglets (C), or 3 separate bile acids each at 0.15% (D; cholate (♦), deoxycholate(▼), glycocholate(•)).** The Stx2e levels in the bacterial culture supernatants were determined by ELISA. Strains that showed a response were divided into two groups: decreased (A,B,C1) or increased (A,B,C2) extracellular Stx2e levels. The three strains used for testing the separate bile salts were strains that showed a pronounced decrease (1, red) or increase (2, green) under the influence of bile acid mixtures. (A-C) Each symbol represents the mean of three independent results (n = 3); D) each symbol represents an independent experiment (n = 3), bars denote the geometric mean. Either the concentration (A-C) or the fold change to the control in mean toxin secretion (D) was plotted for each strain. *, p < 0.05; ***, p < 0.001 vs control.

checked whether these strains carried the genes encoding the QseEF two-component system. The latter allows bacteria to sense and respond to host-derived catecholamines (S9 Table). Interestingly, only one strain (1709) lacked the *qseEF* genes, while they were present in the other strains. We then cultured these STEC strains in the presence of nor- or epi-nephrine and assessed both their growth and concentration of Stx2e in the supernatants. However, our data did not show altered bacterial growth or toxin production for any of the strains (Fig 4).

### Different classes of antibiotics differentially influence Stx2e production

To gain a better insight into the inter-strain variation in response to different in-feed and therapeutic antibiotics in swine, we evaluated extracellular Stx2e levels by the field strains in response to amoxicillin (Amox), tetracycline (Tetra), erythro-mycin (Ery), ciprofloxacin (Cipro), and enrofloxacin (Enro) at 1/4th MIC. For all antibiotics, a significant decrease in growth was observed (Fig 5A). Like previous data [33], the strains showed no change in extracellular Stx2e levels in the presence of amoxicillin, while a significant 2-fold decrease in Stx2e levels was observed for both tetracycline and erythromycin

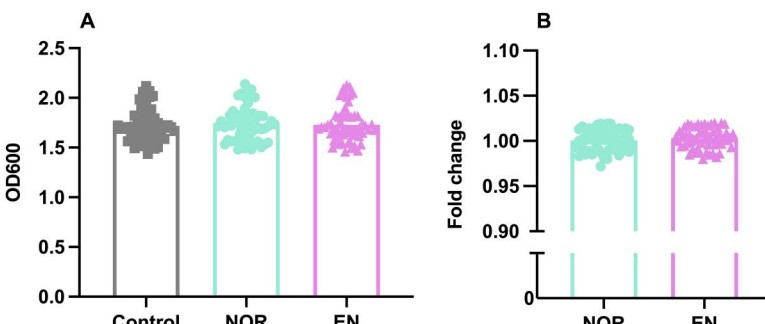

**Fig 4. Influence of nor- and epinephrine on growth and extracellular Stx2e levels – All STEC strains were grown overnight in LB medium with 50 µM norepinephrine (NOR, •) or epinephrine (EN, ▲).** The optical density at 600nm ($OD_{600}$) was used as a measure of bacterial growth (A), and the presence of Stx2e in bacterial culture supernatants was determined by ELISA and presented as a fold change to the control (B). Each symbol represents the mean from three independent experiments, bars denote the geometric mean of all strains.

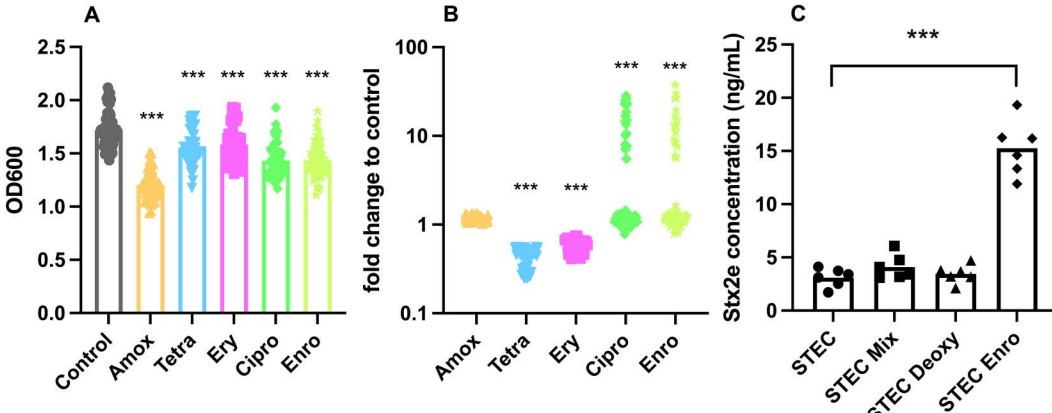

**Fig 5. Influence of antibiotics on STEC growth and extracellular Stx2e levels – STEC strains were grown overnight in LB medium with amoxicillin (Amox), tetracycline (Tetra), erythromycin (Ery), ciprofloxacin (Cipro), and enrofloxacin (Enro) at ¼ the MIC.** (A) The difference in optical density at 600nm ($OD_{600}$) is compared to the control culture. Each symbol represents the geometric mean of three independent experiments, bars denote the geometric mean of all strains. (B) The Stx2e levels in bacterial culture supernatants (1 mL from standardized bacterial cultures at $OD_{600} = 1.0$) was determined by ELISA and the fold change compared to the no-antibiotic control culture is plotted. Each symbol represents the geometric mean of three independent experiments. ***, $p < 0.001$ vs control. (C) Stx2e levels in perfusion outflow of the SISP experiment. STEC were pre-treated with perfusion fluid (STEC), bile acid mixture (STEC + Mix), deoxycholate (STEC + Deoxy) or enrofloxacin (STEC + Enro). Significant differences in Stx2e concentration were observed between the test conditions with enrofloxacin compared to those without. Each symbol represents one animal and represents the mean of two technical replicates. Bars denote the geometric mean of 6 animals. ***, $p < 0.001$ to control.

(Fig 5B). Remarkably, both cipro- and enrofloxacin triggered a substantial increase (7- up to 41-fold) in the extracellular Stx2e levels in the same 17 out of the 57 STEC strains (29.3%) (Fig 5B).

To validate these findings in a more physiologically relevant context, a small intestinal segment perfusion (SISP) assay was performed using one (STEC strain 4080) of these 17 strains. Consistent with the in vitro results, enrofloxacin treated bacteria produced more Stx2e in the SISP assay (Fig 5C), reinforcing the hypothesis that fluoroquinolones increase toxin production and release by inducing the bacterial SOS response. Conversely, the effects of both the bile acid mixture and deoxycholate acid, which significantly decreased extracellular Stx2e levels by this strain *in vitro*, were not observed in the SISP assay.

## Holin contributes to Stx2e release in the presence of antibiotics

To understand why these 17 strains showed increased extracellular Stx2e levels in the presence of fluoroquinolones, the genomic region flanking the stx2e operon of the 57 STEC strains was analysed in more detail. Three different genomic configurations with the *holin* gene downstream of the Stx2e operon were observed in the 57 STEC strains (Fig 6A). The 17 strains showing increased extracellular Stx2e levels upon fluoroquinolone treatment share the same genomic configuration, wherein the Stx2e operon is immediately followed by a *holin* gene (configuration I). This suggests a potential role for this holin in the secretion of Stx2e by STEC strains in response to external stressors. Holin is a protein encoded by certain bacteriophages and plays a crucial role in the lysis of the host bacterial cell, needed for the release of progeny virions. Holins create pores in the inner membrane to facilitate access of endolysins to the outer membrane. We hypothesize that these pores could play a role in the secretion of Stx2e by facilitating transport of the toxin subunits to the periplasmic space. To test this, the lambda red recombination system was used to replace the holin gene with a kanamycin resistance cassette (Kan^R) in two strains (4056 and 1717). These strains were selected as they responded to the presence of fluoroquinolones by significantly increasing their extracellular Stx2e levels. Subsequently, we evaluated extracellular Stx2e levels by these Δholin mutants in response to enrofloxacin. As shown in Fig 6B, deletion of the holin gene abrogated the enrofloxacin-mediated increase in extracellular Stx2e levels in both strains. Additionally, we transformed these deletion mutants and two strains (4055 and 4063) that were previously not inducible by enrofloxacin with a plasmid containing the holin gene under the control of an inducible promotor. Inducing holin expression with IPTG followed by enrofloxacin treatment restored extracellular Stx2e levels in the Δholin mutants to levels similar to the wild-type strains. In the two

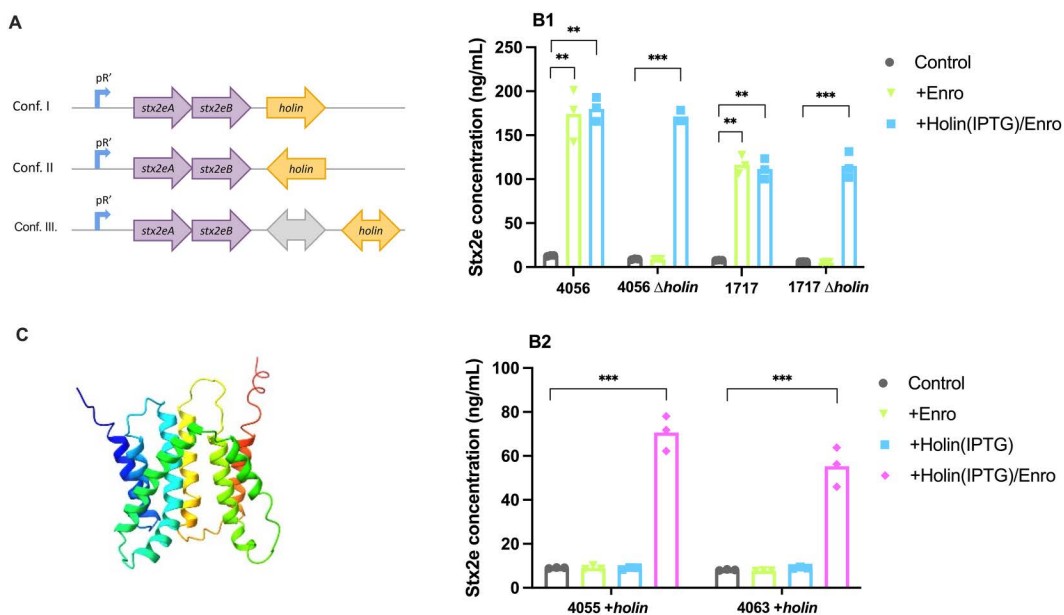

**Fig 6. Influence of enrofloxacin on extracellular Stx2e levels by wild type and Δholin/ ⁺holin mutant STEC strains – (A) In all strains where the holin gene was present, one of these genetic configurations were observed; Stx2e operon followed by I) holin in the same orientation, II) holin in reverse orientation, or III) holin in either orientation but preceded by additional (hypothetical) proteins. All 17 inducible STEC strains in this study belong to configuration (Conf.) I.** The unresponsive strains belong to configuration II or III. (B) Two STEC strains that responded to fluoroquinolones (4056 and 1717) and their Δholin mutants (B1), as well as two STEC strains that did not respond to fluoroquinolones (4055 and 4063) but were transformed with an inducible plasmid containing the holin gene (B2), were grown overnight in LB medium with either no antibiotic (•), enrofloxacin at 1/4^th of the MIC (▼), 1 mM IPTG (■) or enrofloxacin at 1/4^th of the MIC + 1 mM IPTG (♦). The presence of Stx2e in the bacterial culture supernatants was determined by ELISA. The mean concentration of secreted toxin was plotted for each strain. Each symbol represents one independent experiment. **, p<0.01***, p<0.001. (C) The proposed 3D model of the holin protein using Alphafold2.

previously non-inducible strains (4055 and 4063), extracellular Stx2e levels also increased to levels comparable to those of the wild-type strains with inducible holin.

## Discussion

STEC strains play a significant role in the pig production industry due to the increased morbidity, mortality, and reduced productivity associated with infection, resulting in huge economic losses. The Shiga toxin Stx2e is of paramount importance due to its ability to cause ED and bloody diarrhoea in postweaning piglets. Disease severity is correlated with Stx2e concentrations in blood due to more extensive cellular damage and a more pronounced inflammatory response [13,57]. The disease outcome is however highly variable, which could be linked to a potential variation in the ability of different STEC strains to secrete Stx2e. In this study, we investigated both intrinsic differences in Stx2e secretion as well as the influence of several external factors on Stx2e production of 57 genetically distinct porcine STEC field strains.

Previous research using seven STEC strains showed variation in Stx2e production [16–18]. Likewise, the tested strains varied considerably in both their production and secretion levels of Stx2e. Despite other studies using different methods to determine the Stx2e concentration, impeding a direct comparison, most strains are low or moderate secretors, while high secretors are rare. Even though 16 different genomic contexts surrounding the Stx2e operon could be identified, these cannot explain the observed variation in Stx2e production. While we investigated 57 STEC strains, further research will be needed to identify genetic elements that are potentially associated with different Stx2e production levels. To be able to perform GWAS analyses at the highest accuracy, a recent study calculated the number of bacterial strains needed to perform GWAS using a computational pipeline called PowerBacGWAS [58]. To detect moderate to very large effect sizes for genes with a 10% frequency, at least 500–600 high-quality genomes were required, regardless of the bacterial population considered.

In addition to intrinsic elements, external factors can also influence toxin production by STEC strains. The influence of bile acids on pathogenic *E. coli* is multifaceted and encompasses various aspects of strain-specific bacterial physiology, virulence, and host interactions. Here, we observed that bile acids from bovine and porcine origin similarly affected extracellular Stx2e levels in a strain-specific manner with some strains increasing and others decreasing toxin levels. The latter finding partly aligns with previous observations that bile acids suppress certain virulence factors encoded by the Locus of Enterocyte Effacement (LEE) pathogenicity island in one EHEC strain and implies that bile acids might suppress some aspects of STEC virulence [59]. However, the observation that another subset of our strains responded to bile acids with increased extracellular Stx2e levels introduces a novel aspect to our understanding. When testing individual bile acids, only deoxycholate acid altered extracellular Stx2e levels by our STEC strains. This change in Stx2e production only accounted for about 80% of the change observed for the bile acids. In bile, deoxycholate acid is often conjugated with glycine and taurine, which might induce a stronger response [60]. Alternatively, bile acids may act synergistically on Stx2e secretion. Together, the results indicate that STEC strains sense the presence of deoxycholic acids and as such are aware of their arrival in the small intestine. In contrast, in the SISP assay, porcine bile acid mixture and deoxycholic acid did not reduce extracellular Stx2e levels by the tested strain. This might be due to the dynamic nature of bile acid concentrations in the intestinal lumen, interactions with the mucus layer, or potential bacterial adaptations to the host environment. Additionally, the complexity of the intestinal milieu, including competing microbial populations, nutrient availability, and host-bacteria interactions, might modulate the effects of bile acids on toxin production and release. Further research is warranted to elucidate why STEC strains differ in Stx2e production to bile acids and to understand the regulatory and environmental mechanisms that govern toxin production in response to bile acids under in vivo conditions. This might provide valuable insights into the pathogenesis of STEC infections.

In addition to bile acids, bacteria might also encounter catecholamines, which play an important role in host stress responses. These catecholamines might leak into the gut lumen during the weaning period [61]. Previous research

showed that human O157:H7 isolates respond to the catecholamines epinephrine and norepinephrine by expressing Shiga toxins, encoded on their LEE pathogenicity island, and the flagellar regulon [27]. Bacteria sense these molecules via the two-component system QseBC, which activates another two-component system, QseEF. The latter was shown to activate the expression of virulence genes in EHEC. However, the strains tested in this study did not alter their extracellular Stx2e levels in the presence of these catecholamines. This seems to indicate that, in contrast to Shiga toxins from EHEC, Stx2e production and release by porcine STEC is not influenced by catecholamines. Differences in experimental conditions, strain-specific responses, and the regulation of toxin production might explain this discrepancy. Furthermore, quorum-sensing, a process where bacterial communication is mediated through chemical signalling molecules, might play a role in this context as well. Previous research highlighted that quorum sensing can influence the expression of virulence genes in EHEC [27,62]. The quorum sensing signals in the porcine gut may be different from those in human hosts, leading to altered bacterial behaviour and gene expression. Therefore, while the QseBC system is present and potentially functional in the porcine STEC strains, the interplay between catecholamines and quorum sensing signals in regulating Stx2e production and secretion appears to be more complex and context-dependent than in human isolates. Further studies are needed to elucidate these regulatory pathways, the influence of quorum sensing, and their implications for porcine STEC pathogenicity.

Antibiotics were commonly used to treat STEC infections in humans and animals. Their use has become a subject of debate, because antibiotics can trigger the production and release of Shiga toxins, potentially exacerbating disease severity [31,33,34]. Our study confirmed that sub-MIC levels of antibiotics targeting the cell wall (amoxicillin) did not alter Stx2e production, while antibiotics targeting protein synthesis (tetracycline and erythromycin) decreased toxin production. Interestingly, antibiotics targeting DNA replication and in turn also eliciting an SOS response (enrofloxacin and ciprofloxacin) increased extracellular Stx2e levels. While this has been observed before, it was attributed to the activation of the SOS response coupled with cell lysis by endolysins encoded by lytic bacteriophages. In line with previous research, none of the tested STEC strains studied contained a complete bacteriophage or an endolysin gene downstream of the Stx2e operon [35,63]. Despite this absence, enrofloxacin increased extracellular Stx2e levels in a subset of our STEC strains (17/52), which all carry a gene encoding a type I holin directly downstream of the Stx2e operon. Deleting this holin gene abrogated the enrofloxacin-induced increase in extracellular Stx2e, while supplementing this gene in unresponsive strains elevated their extracellular Stx2e levels upon enrofloxacin treatment. This suggests a crucial role for holin in facilitating Stx2e secretion in certain porcine STEC strains. Holins are small proteins encoded by bacteriophages, which play a pivotal role during the lytic cycle. Holins form octo- to decamer homo-oligomers that create a hole or a pore in the inner membrane through which an endolysin is secreted that degrades peptidoglycan in the bacterial cell wall [64]. Based on their structural characteristics, holins are classified into three types: Type I holins have three transmembrane domains (TMDs) with the N-terminus located in the periplasm and the C-terminus in the cytoplasm; Type II holins possess two TMDs with both termini in the cytoplasm; and Type III holins feature a single TMD, with the N-terminus in the cytoplasm and the C-terminus in the periplasm [65]. Interestingly, in *Clostridium difficile* the type I holin TcdE plays a role in the secretion of the enterotoxin TcdA and the cytotoxin TcdB without inducing cell lysis [66,67]. A similar mechanism might be at play in porcine STEC strains. Under steady-state conditions, Stx2e is secreted into the external environment in the absence of bacterial cell lysis [68,69]. However, upon induction by certain antibiotics that prompt the SOS response, production of Stx2e might exceed the capacity of the secretion machinery. In STEC strains harbouring the holin gene downstream of and in the same orientation as the Stx2e operon, the Stx2e A and/or B subunits are most likely transported to the periplasmic space via a pore in the inner membrane formed by the holins. Thus, holins might enable increased transport of Stx2e A and/or B subunits across the inner membrane, leading to an increased presence of Stx2e in the extracellular environment. In support of this, holins can form large, structured pore complexes, allowing the release of large fusion proteins [70]. Since Stx2e is released in the absence of bacterial cell death, these holin pores might be formed in association with other proteins that control the opening of the pore, as such creating a

specific, gated channel that only opens in the presence of the toxin [67]. Alternatively, the toxin itself could for example act as a plug in the pore, preventing the loss of solutes or other proteins from the bacterial cell. Notwithstanding these potential mechanisms, holin forms a pore in the inner cell membrane, allowing for toxin subunit transport to the periplasmic space and subsequent assembly of the holotoxin. However, it is unclear how the toxin is then transported across the outer membrane.

Our findings underscore the risk of exacerbating toxin release by using fluoroquinolones in swine. Despite not being specifically administered to treat swine STEC infections, the use of fluoroquinolones in the European pig industry is still at 1.71 defined daily doses/kg body weight (DDD/kg BW) [71]. Given its ability to induce Stx2e production in certain STEC strains, it is paramount to further reduce the use of these antibiotics but also to invest in more preventive measures such as vaccinations. In addition, the presence of a holin gene in a certain orientation downstream of the Stx2e operon in some STEC strains presents an opportunity for genetic screening in WT populations prior to administering antibiotics. By including the genomic landscape around the Stx2e operon in diagnostic sequencing assays, it should be possible to rapidly identify strains with the ability to elevate Stx2e extracellular levels under certain environmental conditions. This approach could enable early detection and risk assessment of potentially virulent strains, allowing for timely implementation of preventive measures, such as enhanced monitoring and preventing the use of SOS response-inducing antibiotics in the treatment of infections.

In conclusion, our study sheds new light on the intricate mechanisms that impact Stx2e production and release by porcine STEC strains. We observed variations in Stx2e production among strains, with bile acids and antibiotics exerting strain-specific effects on extracellular toxin levels. While deoxycholate emerged as a key modulator of extracellular Stx2e levels, the underlying genetic factors contributing to strain variability remain to be fully elucidated. Our findings underscore the need for further genome sequencing and analysis to identify genetic determinants associated with Stx2e production and release. Moreover, the presence of a *holin* gene in certain strains increases the presence of Stx2e in the extracellular environment, suggesting a potential role in toxin secretion regulation.

## Supporting information

**S1 Fig. Inter-strain difference of genetically distinct STEC strains in Stx2e secretion levels and their Stx2e promoter region.** (A) Maximum-Likelihood (ML) phylogenetic tree of all included STEC strains (n = 52), including strain 1705 as negative control and outgroup (red). Bootstraps <95 are indicated next to branches. Underlined strains were used in an in-depth characterization of the impact of external factors and strains highlighted in green were used in holin deletion experiments. **(B)** Polymorphisms identified in the downstream promoter region as predicted by sigma70pred "scan" (0.92 SVC score), highlighting the -35 box, spacer, -10 TATA box, and Stx2eA codon start.
(TIFF)

**S1 Table. Overview of the tested strains and their characteristics.** The strains were isolated in 2022 and originated from several farms per country. For each strain, the ID, virulence factor genes, hemolytic activity, antimicrobial resistance profile, origin, secretor status, secretion level (in ng/mL), response to bile acids and fluoroquinolones is given. The values represent the mean Stx2e secretion level (in ng/mL) of three independent repeats. AMR, antimicrobial resistance; Amox, amoxicillin; Cefa, cefalexin; Doxy, doxycycline; Tet, tetracycline; Spec, spectinomycin; Sulf, sulfamethoxazole; Flu, flumequine; Kana, kanamycin; Paro, paromomycin; Flor, florfenicol. Stx2e secretor status: low = < 1 ng/mL; moderate = 1–10 ng/mL; high = > 10 ng/mL.
(XLSX)

**S2 Table. Overview of the minimal inhibitory concentration (MIC) of the tested STEC strains for the given antibiotics.** EM, erythromycin; TC, tetracycline; AC, amoxicillin; CIP, ciprofloxacin; ENRO, enrofloxacin.
(XLSX)

**S3 Table. Overview of the genome sequence quality of the tested strains.** For each strain, median read length, number of reads, total bases, median coverage, completeness, contamination, GC content, genome size, contigs, longest contig, coding density and predicted number of genes is given.
(XLSX)

**S4 Table. Grouping of strains for the GWAS analysis.** Strains were divided in two groups based on their Stx2e secretion level. In group 1, strains were divided in low (0) and moderate/high (1) Stx2e secreting strains, while in group 2 strains were divided in low/moderate (0) and high secreting strains (1).
(XLSX)

**S5 Table. Overview of the GWAS results performed on group 1.** See S4 Table for group definitions and strain assignment.
(XLSX)

**S6 Table. Overview of the GWAS results performed on group 2.** See S4 Table for group definitions and strain assignment.
(XLSX)

**S7 Table. Overview of Stx2e production and secretion by the tested strains.** Stx2e production and secretion were measured by ELISA in three independent experiments.
(XLSX)

**S8 Table. Overview of the genes checked for the STEC strains.** Gene name, gene ID and accession number are listed.
(XLSX)

**S9 Table. Overview of the similarity for each gene (S8 Table) for the tested STEC strains.** --mincov and --minid were run both at 80% or 60% as specified in the table.
(XLSX)

## Acknowledgments

We thank Simon Brabant and Charlotte Helsmoortel for their technical assistance in strain collection, isolation, and characterisation and for helping to determine the MIC values against the tested antibiotics. We acknowledge prof. dr. Stijn Schauvlieghe and Charlotte Cuypers (Department of Large Animal Medicine, Faculty of Veterinary Medicine, Ghent University) for their assistance with the anesthesia needed for the SISP assay.

## Author contributions

**Conceptualization:** Sander Van hoorde, Nick Vereecke, Bert Devriendt.

**Data curation:** Sander Van hoorde.

**Formal analysis:** Sander Van hoorde, Nick Vereecke, Emma Vanbeylen, Emma Van Denberghe, Bert Devriendt.

**Funding acquisition:** Daniel Sperling, Eric Cox, Bert Devriendt.

**Investigation:** Sander Van hoorde.

**Methodology:** Sander Van hoorde, Nick Vereecke, Emma Vanbeylen, Emma Van Denberghe.

**Project administration:** Daniel Sperling, Eric Cox, Bert Devriendt.

**Resources:** Nick Vereecke, Daniel Sperling, Xiaohua He, Eric Cox, Bert Devriendt.

**Supervision:** Daniel Sperling, Eric Cox, Bert Devriendt.

**Validation:** Sander Van hoorde.

**Visualization:** Sander Van hoorde, Nick Vereecke.

**Writing – original draft:** Sander Van hoorde, Nick Vereecke, Bert Devriendt.

**Writing – review & editing:** Sander Van hoorde, Nick Vereecke, Daniel Sperling, Xiaohua He, Emma Vanbeylen, Emma Van Denberghe, Eric Cox, Bert Devriendt.

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
