## [Decision Letter · Decision Letter 0]

5 May 2025

External Factors Influence Intrinsic Differences in Stx2e Secretion by Porcine Shiga Toxin-Producing Escherichia coli Strains

PLOS Pathogens

Thank you for submitting your manuscript to PLOS Pathogens. The manuscript was carefully and exhaustively reviewed by two external reviewers. Both found merit in the manuscript with clear relevance to porcine STEC pathogenesis and toxin regulation. However, both reviewers identified several drawbacks in the presentation and explanation of the data used to sustain the main conclusions of the work. Therefore, after careful consideration, we feel that as it currently stands, the manuscript does not fully meet PLOS Pathogens's publication criteria. However, if you are able to carefully and clearly address all of the major and minor points raised by the two reviewers, we will be happy to consider a revised version of the manuscript.

If you are able to address these issues, please submit your revised manuscript within 60 days Jul 04 2025 11:59PM. If you will need more time than this to complete your revisions, please reply to this message or contact the journal office at plospathogens@plos.org. **Please include the following items when submitting your revised manuscript:**

*** An unmarked version of your revised paper without tracked changes. You should upload this as a separate file labeled 'Manuscript'.**

We look forward to receiving your revised manuscript.

Kind regards,

Chuck S. Farah, PhD

Academic Editor

PLOS Pathogens

David Skurnik

Section Editor

PLOS Pathogens

Editor-in-Chief

PLOS Pathogens

orcid.org/0000-0003-2946-9497

Michael Malim

Editor-in-Chief

orcid.org/0000-0002-7699-2064

**Journal Requirements:**

At this stage, the following Authors/Authors require contributions: Sander Van hoorde, Nick Vereecke, Daniel Sperling, Xiaohua He, Emma Vanbeylen, Emma Van Denberghe, Eric Cox, and Bert Devriendt. Please ensure that the full contributions of each author are acknowledged in the "Add/Edit/Remove Authors" section of our submission form.

https://journals.plos.org/plospathogens/s/submission-guidelines#loc-parts-of-a-submission

5) We have noticed that you have uploaded Supporting Information files, but you have not included a list of legends. Please add a full list of legends for your Supporting Information files after the references list.

7) Please ensure that the funders and grant numbers match between the Financial Disclosure field and the Funding Information tab in your submission form. Note that the funders must be provided in the same order in both places as well. Currently, the order of the funders is different in both places.

**Reviewers' Comments:**

Reviewer's Responses to Questions

**Part I - Summary**

Reviewer #1: Production and secretion of shiga-like toxin stx2b in 57 porcine STEC isolates were studied. The main conclusions are that variations in secretion of toxin in response to antibiotic and environmental stressors are in some isolates dependent on presence of a holin gene downstream of the stxAB genes. Deletion of the holin gene convincingly showed that production/secretion of stx2b was reduced and in complemented isolates the production/secretion was increased.

The authors also showed that deoxycholic acid affected different isolates in different ways, either by up or downregulation of secretion while catecholamines had no effect. Addition of submic antibiotics ciprofloxacine and enrofloxacine also induced stx secretion in a subset of isolates. The tested substances were chosen due to reported ability to induce the SOS response.

The paper is nicely written and very interesting, but I do have some concerns and questions.

Reviewer #2: This study addresses an important topic related to Shiga toxin-producing E. coli (STEC) in pigs, focusing on the factors influencing stx2e toxin production and secretion. The work combines phenotypic assays with genomic data, which is a strength. The topic is highly relevant for both understanding STEC pathogenesis and informing potential therapeutic strategies.

However, the study currently has several weaknesses that limit its impact. The manuscript does not fully utilize the available long-read sequencing data, restricting the genomic analysis to 10 kb flanking regions without deeper exploration of prophage structures, plasmid associations, or structural variation. In addition, there are issues with clarity and consistency throughout the manuscript, particularly in the description of methods, results, and figure presentations.

Despite these issues, the study has the potential to make a meaningful contribution to the field if the authors address the concerns raised, clarify and expand their genomic analyses, and improve the organization and clarity of the manuscript.

**Part II – Major Issues: Key Experiments Required for Acceptance**

Reviewer #1: The authors conclude that secretion of stx2b is highly variable but almost all produced stx2b is also secreted (fig 1 and suppl table 7), hence the results are not linked to secretion but to production and thereby probably regulation of expression.

In Fig 5A the growth at sub MIC levels of the antibiotics are analyzed and all are significantly different i.e downregulated compared to the control as indicated with three stars, this is not apparent from the data in the figure, is it correct and if so, a downregulation in growth might affect other factors than what the authors intended to analyze.

Required experiment: Could the authors complement the studies with analysis of toxin production over time and growth phase with and without sub-MIC antibiotics.

Also in fig 5B cipro and enro gives two different subpopulations, how was the statistical analysis performed to take this into account since apparently one subpopulation has no change in response to these two antibiotics. In this figure the same 17 isolates that were upregulated by cipro apparently had a significant downregulation in response to tetra and ery but where they analysed separately from the other isolates and did the others not change? Please clarify.

The authors continue to analyze the 17 isolates that had increased production/secretion in response to cipro. A holin gene was found downstream in all isolates but that was also found in totally 38 isolates, please clarify the names of the 17 isolates and indicate them in fig 2. Also please clarify if there is any correlation between the response to cipro and deoxycholate upregulation/downregulation in these 17 isolates as well as in all isolates.

The holin mutant and complemented mutant experiments are convincing but since secretion and expression are so similar in these isolates it might be involved in transcriptional or translational regulation rather than secretion.

required analysis: The study should be complemented with an analysis of the promoter regions of stxAB for all isolates to determine if there are differences in binding sites for transcription factors that correlate with responses to deoxyxholate and cipro. In addition, gene expression analysis of stx2b in presence and absence of holin can be analysed to determine if holin affects gene expression of stx2b

Reviewer #2: Expanded genomic analysis using long-read data: The current analysis is limited to 10 kb flanking regions around the stx2e genes. A more comprehensive characterization using the long-read assemblies is necessary. Specifically, the authors should clarify whether stx2e is plasmid- or chromosome-encoded, perform detailed prophage characterization, and explore structural variations or broader synteny surrounding the stx2e operon.

Clarification and complete description of genomic methods: The Materials and Methods section should include a clear description of the genome assembly, quality control procedures, SNP calling, and phylogenetic tree construction. The authors should specify the software versions, settings used, and explain whether the phylogenetic tree was based on SNPs from the core genome and how SNPs were defined.

Improved data presentation and figure clarity: Figures (particularly Figures 1 and 2) need revision to clearly show strain IDs, secretion levels, and genomic organizations. Categorization of strains into secretion levels (low, moderate, high) should be indicated consistently. Legends should be updated to match the figure contents, and visual clarity should be improved, for example by using standard tree layouts and adjusting color usage in the clinker plot.

**Part III – Minor Issues: Editorial and Data Presentation Modifications**

Reviewer #1: Line 103: remove ref 22

Line 109: ref 43? Should it be renumbered to 27? Or is one reference missing? Ref 43 is a python tool used to determine flanking regions. Check numbering of all references after addressing ref 43.

Fig 5B, there are no bars in the figure.

Reviewer #2: • References are not listed in order throughout the manuscript.

• Some references are missing and should be added where relevant.

• Line 415 refers to the wrong table; it should refer to Table S7.

• Table S2 is not referred to in the text; please update.

• The GWAS section should be removed, as it is not properly integrated or supported.

• Was AMR (antimicrobial resistance) genomically analyzed in addition to experimental testing? Please clarify. Line 408 indicates this but no analysis is shown.

• Analysis of fimbriae/adhesion molecules: many strains are negative for adhesion molecules in Table S1. Were factors beyond F4 and F18 checked?

• Why focus only on F4 and F18 CFs in Table 1? Could other adhesion molecules be relevant?

• The number of strains varies between different analyses (58, 57, 53, 52). Please clarify these differences where relevant.

• Clarify what is meant by "genomic regions" — does it refer to upstream, downstream, or all flanking regions shown in Figure 2?

• Checking for QseEF: no information is provided on methodology.

• Table S1: Include the year of isolation. Indicate whether isolates came from one farm or several farms per country.

• Table S3: Please add a strain ID column for easier cross-referencing.

• Table S4 (GWAS output): Define Group 1 and Group 2 properly.

• Table S8: Include a description of contents.

• Table S9: Clarify the coverage and identity thresholds used in abricate analysis; the text states 80% coverage and 60% identity, but Table S9 lists 80% for both.

• Materials and Methods: Specify the number of biological and technical replicates, e.g., for ELISA experiments.

• How were the groupings for level of secretion defined — based on cytotoxicity or previous publications? Clarify.

• Why only investigate stx2e? Clarify the scientific motivation.

• Figure 1: The number of strains does not match the legend. Strain IDs are missing. Consider making the bar horizontal and indicating strain IDs if possible.

• Figure 2:

A more standard phylogenetic tree layout (rectangular or slanted branches) would improve clarity, especially for readers comparing branch lengths and tip relationships.

Bootstrap values below 95% are currently plotted along the branches; place them next to internal nodes to avoid confusion and update the figure legend accordingly.

The clinker figure uses many colors which are not described; this makes interpretation difficult. Many genes of interest and flanking CDSs downstream of holin are colored purple, making them hard to distinguish. Remove unnecessary colors.

Add a scale ruler on top of the clinker figure to indicate the region length instead of using a scale bar.

Use colors to highlight stx2e genes carrying non-synonymous mutations instead of writing mutations as text, which is difficult to see.

Describe E46* clearly in the figure legend.

Highlight which strains are low, moderate, and high secretors in the clinker figure so that secretion levels can be cross-referenced with Figure 1.

In the updated Figure 2 with a bar graph, color the bars based on secretion level (low, mid, high).

It is mentioned in Figure 6 that different holin gene conformations exist, but this is not visible in Figure 2. Clarify differences between strains, e.g., strain 4055 and 4056.

• Figure 3A–C: Clarify whether squares and circles represent different conditions.

• In bile component secretion assays (13 strains), either adjust the figure to include strain IDs or refer clearly to a supplementary table that lists concentration and outcome.

• Figure 4: Recommend moving to Supplementary Figures for better flow. Also, the figure legend does not describe Figure 4B (fold change).

• Figure 5B: Bars are missing; please clarify or correct.

• Line 371: The text states "three non-silent SNPs," but five mutations (D175E, T207A, P289L, T296K, S313P) are listed. Clarify and correct either the number of SNPs or the description of the mutations.

• Recommend including a multiple sequence alignment (MSA) of the stx2e sequences as a supplementary figure to clearly show the non-synonymous SNPs.

• Specify how many strains showed increased versus decreased stx2e secretion when tested with different bile components.

• In the genomic context section, specify which genomic context is referred to and clarify if clusters are based on phylogenetic clades.

• Clarify how genomic regions were defined (e.g., 10,000 bp each side or total 10,000 bp?).

• Did you also look at larger genomic contexts beyond 10,000 bp? If so, please discuss findings.

• In the abstract, “potential new therapeutics” are mentioned — please elaborate briefly in the Discussion on what kinds of therapeutics the authors envision.

• Recommend shortening the Discussion and removing or minimizing references to GWAS results unless strongly supported.

PLOS authors have the option to publish the peer review history of their article (what does this mean? ). If published, this will include your full peer review and any attached files.

**Do you want your identity to be public for this peer review?** For information about this choice, including consent withdrawal, please see our Privacy Policy .

Reviewer #1: No

Reviewer #2: No

**Figure resubmission:**

**Reproducibility:**



---

## [Decision Letter · Decision Letter 1]

14 Aug 2025

PPATHOGENS-D-25-00599R1

External Factors Influence Intrinsic Differences in Stx2e Secretion by Porcine Shiga Toxin-Producing Escherichia coli Strains

PLOS Pathogens

Dear Dr. Devriendt,

Thank you for submitting your revised manuscript to PLOS Pathogens. The revised version was reviewed by teh two original external referees. While one was satisfied with the changes made, the other referee still has pointed out some issues that  need to be addressed. Normally, we only allow one round of revision to fully meet PLOS Pathogens's publication criteria. Aster carefull consideration, we invite you to submit a revised version of the manuscript to carefully addresses these outstanding issues.

Please submit your revised manuscript within 30 days Oct 13 2025 11:59PM. If you will need more time than this to complete your revisions, please reply to this message or contact the journal office at plospathogens@plos.org. Please include the following items when submitting your revised manuscript:

We look forward to receiving your revised manuscript.

Kind regards,

Chuck S. Farah, PhD

Academic Editor

PLOS Pathogens

David Skurnik

Section Editor

PLOS Pathogens

Sumita Bhaduri-McIntosh

Editor-in-Chief

PLOS Pathogens

orcid.org/0000-0003-2946-9497

Michael Malim

Editor-in-Chief

PLOS Pathogens

orcid.org/0000-0002-7699-2064

**Journal Requirements:**

**Reviewers' Comments:**

Reviewer's Responses to Questions

PLOS authors have the option to publish the peer review history of their article (what does this mean? ). If published, this will include your full peer review and any attached files.

**Do you want your identity to be public for this peer review?** For information about this choice, including consent withdrawal, please see our Privacy Policy .

Reviewer #1: No

Reviewer #2: No

**Part I - Summary**

Reviewer #1: The ms has improved and is still very interesting in my opinion but there are still some items that needs correction or clarification.

**Part II – Major Issues: Key Experiments Required for Acceptance**

Reviewer #1: The levels of production/expression and secretion are nearly identical in fig1, hence all expressed stx2b are also secreted. But the authors argue that secretion itself is mediated by holin causing more holes or pores in the outer membrane thereby increasing secretion out of the bacterium. But check for example 1717 in presence of Enro, the induction of secreted toxin is increased from 7 to 116 ng and since all expressed/assembled toxin is also secreted through the outer membrane in normal conditions (as proved in fig 1) holin plus enro somehow induce increased levels of mature stx2bAB holotoxin, but only in presence of enro (by SOS responses). Since holin is located in the inner membrane and I assume stx2b is assembled in the periplasm maybe holin helps the A and B subunit to reach the periplasm. I suggest another term, why not use “stx2b assembly” or “production of mature holotoxin” I think using secretion and even secretion levels are not backed up by your results because more toxin is produced (and hence released).

Figure 2 has improved substantial with the changes but mark the 17 isolates that show the enro response, I cannot see the reason not to and it helps the reader.

There is a discrepancy in strains used that needs to be addressed Fig S1 and fig 2 mark holin analysed strains in green, i.e 4056, 4064 and 4057 and 1717, these are also mentioned in the text line 338 (4056, 4064, 4057 Δholin, and 1717 Δholin)

Then compare line 563-564: To test this, the lambda red recombination system was used to replace the holin gene with a kanamycin resistance cassette (KanR) in two strains (4056 and 1717).

Line 568-571: Additionally, we transformed these deletion mutants and two strains (4055 and 4063) that were previously not inducible by enrofloxacin with a plasmid containing the holin gene under the control of an inducible promotor. Hence different isolates are mentioned

4056 is responsive to enro according to table S1 but appears genetically identical to 4064 and 4055 who are unresponsive but do have a holin gene at the same place. Why would complementation of holin work for 4064 and 4055? And what isolates were used? only 1717 is consistent.

The new analysis of the promoter region is nice but only include -35 and -10 regions, transcription factors probably bind upstream of -35 so can the region be expanded 5´

**Part III – Minor Issues: Editorial and Data Presentation Modifications**

Reviewer #1: I disagree with the discussion about PCR assays for holin presence since it does not say anything, holin is there also in unresponsive isolates. If there is a difference in sequences or holin promoter region this must be addressed and explained in eg fig2

**Figure resubmission:**
---

## [Decision Letter · Decision Letter 2]

12 Oct 2025

Dear Dr. Devriendt,

We are pleased to inform you that your manuscript 'External Factors Influence Intrinsic Differences in Stx2e Production by Porcine Shiga Toxin-Producing Escherichia coli Strains' has been provisionally accepted for publication in PLOS Pathogens.

Best regards,

Chuck S. Farah, PhD

Academic Editor

PLOS Pathogens

David Skurnik

Section Editor

PLOS Pathogens

Sumita Bhaduri-McIntosh

Editor-in-Chief

PLOS Pathogens

orcid.org/0000-0003-2946-9497

Michael Malim

Editor-in-Chief

PLOS Pathogens

orcid.org/0000-0002-7699-2064

Reviewer Comments (if any, and for reference):

Reviewer's Responses to Questions

**Part I - Summary**

Reviewer #1: I am satisfied with the improvements

**Part II – Major Issues: Key Experiments Required for Acceptance**

Reviewer #1: (No Response)

**Part III – Minor Issues: Editorial and Data Presentation Modifications**

Reviewer #1: (No Response)

PLOS authors have the option to publish the peer review history of their article (what does this mean? ). If published, this will include your full peer review and any attached files.

**Do you want your identity to be public for this peer review?** For information about this choice, including consent withdrawal, please see our Privacy Policy .

Reviewer #1: No

---

## [Editor Report · Acceptance letter]

Dear Dr. Devriendt,

We are delighted to inform you that your manuscript, "External Factors Influence Intrinsic Differences in Stx2e Production by Porcine Shiga Toxin-Producing Escherichia coli Strains," has been formally accepted for publication in PLOS Pathogens.

Best regards,

Sumita Bhaduri-McIntosh

Editor-in-Chief

PLOS Pathogens

orcid.org/0000-0003-2946-9497

Michael Malim

Editor-in-Chief

PLOS Pathogens

orcid.org/0000-0002-7699-2064